# Long-Term Corn–Soybean Rotation and Soil Fertilization: Impacts on Yield and Agronomic Traits

Ming Yuan [1], Yingdong Bi [2,*], Dongwei Han [1], Ling Wang [2], Lianxia Wang [1], Chao Fan [2], Di Zhang [1], Zhen Wang [1], Wenwei Liang [2], Zhijia Zhu [1], Yuehui Liu [3], Wei Li [2], Haoyue Sun [1], Miao Liu [2], Jianxin Liu [2], Junqiang Wang [1], Bo Ma [1], Shufeng Di [2], Guang Yang [2] and Yongcai Lai [2]

[1]   Qiqihar Branch of Heilongjiang Academy of Agricultural Sciences, Qiqihar 161006, China
[2]   Institute of Crop Cultivation and Tillage, Heilongjiang Academy of Agricultural Sciences, Harbin 150028, China
[3]   Agricultural Technology Extension Center of Nenjiang, Nenjiang 161499, China
*   Correspondence: yingdongbi@haas.cn; Tel.: +86-451-51127890

**Abstract:** Although crop rotations have been widely shown as an effective approach for improving yield or soil quality in the long term, the relationship between crop rotations and quality traits of crop or biochar-based fertilization is still unclear. To address this, we conducted a long-term field experiment in the Heilongjiang province of China to investigate the effects of crop rotation and biochar-based fertilizer application on the crop yield, soil parameters, crop quality, and agronomic traits in 2014–2020. The effects of rotations on crop production and soil properties were evaluated with the average yield variability during the 7 years of this study. Our results showed that the diversified corn and soybean rotations had a significant positive effect on average crop yield compared with their monocultures. The corn yield was enhanced by 0.6 Mg ha$^{-1}$ (5.4%) in the corn–soybean–corn (CSC) crop sequence compared with monoculture corn. Similarly, soybean yield was enhanced by 0.21 Mg ha$^{-1}$ (9.7%) in the soybean–corn–corn (SCC) crop sequence compared with monoculture soybean. However, a negative effect of crop rotations was detected on the protein content of soybean compared with the monoculture soybean, while a positive effect was detected on oil content. Additionally, no differences were detected in crop yield between biochar-based fertilization and mineral fertilization treatments, but a significant positive effect of biochar-based fertilization was observed for any crop on both protein and oil content. A significant effect of crop rotation was found on the percentage of total soil N (TN), available soil N (AN), and available soil K (AK) content. The SSS crop sequence treatment illustrated the highest TN values at 0.18%. The CCC crop sequence treatment increased AN and AK content by 9.1% and 7.8%, respectively, compared with SSS ($p < 0.05$). We conclude that crop rotations increase crop yield and biochar-based fertilizer application, improving crop quality traits in the long term. Thus, the addition of biochar-based fertilizer could efficiently enhance the yield and quality of crop in the rotation cropping system. The findings of this study may provide useful information for designing sustainable cropping systems based on rotations.

**Keywords:** crop rotation; corn; soybean; yield; agronomic traits

## 1. Introduction

Crop rotation is an agricultural management practice of sequentially growing different crop species on the same land successively [1]. It was considered a useful tool for breaking the life cycles of weed or pest and enhancing soil fertility [2,3]. Crop rotation is also an important strategy for the design and realization of sustainable agricultural systems due to its additional benefits [4–6]. Corn and soybean are both economically important food crops whose production accounts for 30% and 50% of the country's total output, respectively, in Northeast China and which are often rotated with different crop every other growing season. However, evidence indicates that crop yield often declines under continuous monoculture [7–9]. A field study in southeastern Kansas found that the yield of soybean



in rotation with another crop was greater than that of continuous monocrop [10,11]. The rotation of corn and soybean and its beneficial effects have been well studied, and this rotation system has been proven to be economically significant compared with continuous monoculture [12,13]. M.J. Kazula et al., (2018) reported that the yield of corn and soybean increased within the context of rotation. Corn yield in corn–soybean–wheat (CSW) rotation was 15% and 8% greater than that in corn–corn (CC) and corn–soybean (CS) rotation, respectively [14]. Soybean yields in rotations were found to be more than 40% greater than continuous monocrop. In addition, Lindsay A. Chamberlain showed that rotation cropping systems were key to achieving high yields for both corn and soybean [15].

Numerous studies have been carried out to identify the beneficial effects of rotation on soil properties and crop yield over time [16,17]. The positive effects of crop rotation on soil organic carbon have been observed, especially in crop rotations with legumes with the addition of organic matter and nitrogen (N) to the soil [18,19]. Green's research showed that optimum N fertilizer rates for corn following soybean were lower than following corn, and they postulated that this was due to N fixation by soybean or change patterns of mineralization [20]. Martens et al., (2006) reported that the full amount of N fertilizer may not be required for soil following soybean to achieve optimum yield [21]. Many studies have been performed to determine the optimum rates for mineral fertilizers application during the last few decades and to refine recommendations for their application in crop rations [22,23]. Soybean could also act as an N sink and may help to reduce soil N leaching since soybean in rotation systems could utilize soil or fertilizer N and take advantage of other rotational benefits [24]. The applications of biennial fertilizer were often used preceding corn to meet the fertility needs of maize and soybean in high-yield environments [22]. However, these benefits of application of organic amendments to agricultural soil depend heavily on the quantity and nature of organic wastes [25]. Organic amendments also play an important role in mitigating climate change through soil carbon sequestration, which depends on the rates or the frequency of application. Sustainable fertilizer management developed through integration with many sources of organic nutrients could improve soil fertility and quality [26–28]. Biochar0based fertilizer application has generally proved more effective in increasing yield and presented with better environmental performance [29,30]. However, to date, the effect of biochar-based fertilizer on the yield and agronomic traits of crop and its relationship with corn–soybean rotation are rarely documented.

Therefore, the objectives of our study were (i) to determine the role of corn–soybean rotations and continuous monocropping in shaping the yield and agronomic traits of crop in China's major grain producing areas; (ii) to compare soil properties and yields in continuous and rotated corn–soybean production systems; and (ii) to evaluate the effects of fertilizer treatments on the yield and quality traits of crop in long-term experiments.

## 2. Materials and Methods

### 2.1. Description of Experimental Site

A long-term crop rotation experiment was initiated at the Qiqihar branch of the Heilongjiang Academy of Agricultural Sciences, Fularji District research base, in 2014. The agricultural environment of the experimental site (47°15′ N, 123°40′ E) was a semi-arid climate with an average annual rainfall of 400–550 mm, most of which falls during May to September. The average temperature in summer is 20.6 °C. The soil at the site is identified as Alkaline meadow soil, which is classified as chernozems soil type according to FAO soil groups. Monthly precipitation and mean temperatures from 2014 to 2020 are shown in Table 1.

**Table 1.** Meteorological parameters for the experimental site (Qiqihar, Heilongjiang province).

| Year | Temperatures/ Precipitation | May | | | Jun | | | Jul | | | Aug | | | Sep | | |
|---|---|---|---|---|---|---|---|---|---|---|---|---|---|---|---|---|
| | | Early | Middle | Late | Early | Middle | Late | Early | Middle | Late | Early | Middle | Late | Early | Middle | Late |
| 2014 | Temperature (°C) | 11.5 | 13.7 | 18.4 | 22.1 | 22.9 | 24.8 | 23 | 22.1 | 22.6 | 22.3 | 20.1 | 21.7 | 16.9 | 15.5 | 11 |
| | Precipitation (mm) | 0 | 47.6 | 12.3 | 19.5 | 1.1 | 2.1 | 53.9 | 63.5 | 22.9 | 10.1 | 77.9 | 39.6 | 45.6 | 1.9 | 16.4 |
| 2015 | Temperature (°C) | 9.7 | 11.7 | 18.4 | 17.5 | 22.4 | 24 | 23.8 | 23.9 | 23.2 | 23.5 | 22.9 | 20.1 | 17 | 16.5 | 10.9 |
| | Precipitation (mm) | 4.9 | 45.6 | 0.06 | 36.4 | 42.2 | 35.5 | 0 | 34.9 | 74.7 | 29.1 | 33.4 | 19 | 5.4 | 0 | 49.7 |
| 2016 | Temperature (°C) | 14.3 | 17.2 | 17 | 19.6 | 18.9 | 21.4 | 25.6 | 23.9 | 24.6 | 23.7 | 23.4 | 19.2 | 18.9 | 15.9 | 13 |
| | Precipitation (mm) | 20.3 | 1 | 22.9 | 6.4 | 37.1 | 75.4 | 0.5 | 8.2 | 2.8 | 17.5 | 2.8 | 13.6 | 36.9 | 8.2 | 28.6 |
| 2017 | Temperature (°C) | 16 | 18.2 | 15.7 | 16.7 | 21.6 | 25.5 | 27.7 | 26.3 | 23.1 | 23.2 | 23.9 | 18.1 | 17.3 | 14 | 12.6 |
| | Precipitation (mm) | 15.9 | 0.2 | 1.5 | 12.8 | 27.3 | 9.3 | 9.7 | 7.3 | 16.1 | 27.1 | 49.9 | 2.3 | 55 | 7.2 | 5.4 |
| 2018 | Temperature (°C) | 13.4 | 18.7 | 16.6 | 22.7 | 19.7 | 21.6 | 23.1 | 25.4 | 24.8 | 22.6 | 21.4 | 20.1 | 15.7 | 16.8 | 14.2 |
| | Precipitation (mm) | 0 | 0.1 | 15.2 | 10.8 | 46.3 | 43.5 | 66.2 | 61.6 | 73.6 | 24.3 | 21.6 | 42.1 | 58.7 | 8.9 | 9.6 |
| 2019 | Temperature (°C) | 15 | 15.5 | 16 | 18.6 | 20.8 | 20.8 | 21.7 | 24.8 | 23.1 | 21.7 | 20.2 | 19.1 | 20.7 | 14.2 | 16.2 |
| | Precipitation (mm) | 25.7 | 0.2 | 20 | 21.4 | 27.7 | 44.2 | 23.5 | 88.7 | 91.2 | 95.2 | 80.4 | 3.4 | 3.2 | 2.1 | 1.2 |
| 2020 | Temperature (°C) | 14.4 | 15.5 | 17.7 | 19 | 20.1 | 19.6 | 24.3 | 27.1 | 27 | 22.1 | 20.7 | 19.9 | 17.6 | 14.8 | 13.7 |
| | Precipitation (mm) | 1.9 | 7.2 | 20.6 | 23 | 42.7 | 41.3 | 9.4 | 0.1 | 37.3 | 83.8 | 53.3 | 35.2 | 16.4 | 24.4 | 2.5 |
| Averages | Temperature (°C) | 13.5 | 15.8 | 17.1 | 19.5 | 20.9 | 22.5 | 24.2 | 24.8 | 24.1 | 22.7 | 21.8 | 19.7 | 17.7 | 15.4 | 13.1 |
| | Precipitation (mm) | 9.8 | 14.6 | 13.2 | 18.6 | 32.1 | 35.9 | 23.3 | 37.8 | 45.5 | 41.0 | 45.6 | 22.2 | 31.6 | 7.5 | 16.2 |

### 2.2. Experimental Design and Treatments

The long-term field trial was established in 2014 to investigate the effects of different crop rotation combinations and soil fertilization on yield and agronomic traits of corn and soybean. A split plot designed experiment was arranged with three replicates in the field, where crop rotation (continuous corn (CCC), continuous soybean (SSS), corn–soybean–corn (CSC), soybean–corn–corn (SCC)) was the main plot, and soil fertilization treatment (biochar-based fertilization (BF) and mineral fertilization (MF)) was the split plot. Fertilization was carried out in spring. The treatments of soil fertilization were defined as follows: biochar-based fertilization (BF, N-$P_2O_5$-$K_2O$: 105-75-75, 255 kg/ha with 750 kg/ha biochar for corn; N-$P_2O_5$-$K_2O$: 38.5-60-45, 143.5 kg/ha with 750 kg/ha biochar for soybean) and mineral fertilization (MF, N-$P_2O_5$-$K_2O$: 150-75-75, 300 kg/ha for corn and N-$P_2O_5$-$K_2O$: 55-60-45,160 kg/ha for soybean) as control. Each plot was approximately 6.5 m wide (10 rows, 65 cm row spacing) and 20 m long. Typical corn and soybean varieties were used according to the local recommendations each year, which are corn cultivar (Nendan 19) and soybean cultivar (Nenfeng 16).

### 2.3. Field Agronomic Practices

For this 7-year maize–soybean rotation, the moldboard plowing treatment included fall moldboard plowing (20–25 cm deep) after corn, spring disking (7.5–10 cm deep), and field cultivating; fall chisel plowing (30–35 cm) was carried out after soybean, followed by spring disking and field cultivation. The fall moldboard plowing (30 cm deep) was carried out with crop straws returning to the field after harvesting, followed by field cultivating (10 cm deep) prior to sowing. Throughout the 7-year study, the corn cultivar 'Nendan 19' was seeded at a rate of 30 kg ha$^{-1}$, and uniform plant population was maintained at a density of 60,000 ha$^{-1}$ by thinning one week after emergence. The soybean cultivar 'Nenfeng 16'

was seeded at a rate of 60 kg ha$^{-1}$, and uniform plant population was maintained at a density of 200,000 ha$^{-1}$ by thinning one week after emergence. Sowing dates ranged from 28 April to 15 May. Herbicide (acetochlor and thifensulfuron) were applied for the need of vegetation control before sowing. Hand weeding post-emergence was used to control the dominant weeds given inadequate pre-emergence control. All agronomic practices have been maintained uniformly for all plots.

### 2.4. Crop Measurements and Soil Sampling

Plant height (PH) was measured 1 week before the harvesting of three selected plants randomly in the center of each plot. Number of pods per plant (NP), number of grains per plant (NG), hundred grain weight (HW), length of corn ear (LE), and weight of corn ear (WE) were determined for each plot. The central 4 rows of each 6-row plot were mechanically harvested for yield determination using a plot harvester. Grain dry weight of maize and soybean crops were recorded for biological yield. Seed protein, oil, and fatty acid analysis was conducted using a near-infrared (NIR) reflectance diode array feed analyzer (Perten, Springfield, IL, USA) for each crop plot. Updating of the calibration curve was performed according to the AOAC method. Three soil samples were collected at a 15 cm depth in each plot with stainless-steel cylinder augers (50.46 mm dia. $\times$ 50 mm length). after harvesting. The soil samples were air-dried after removing any vegetation remains, then ground to a fine powder which was passed through a 2 mm sieve and kept in zip-lock plastic bags until used for analyses. The soil samples were analyzed at the Center for Quality Supervision, Inspection and testing of cereals and their products, Ministry of Agriculture and rural areas, Harbin; the soil pH (1:2.5 soil:water) was determined using an FE28-Standard composite electrode (Mettler Toledo) to determine SOC contents, available soil N, available soil P, available soil K, total soil N, total soil P, and total soil K. Carbonates were washed from the soil using hydrochloric acid, and total SOC and N contents were then determined using a VarioEL CHN elemental analyzer (Heraeus Elementar Vario EL, Hanau, Germany). The contents of total soil P (TP), available soil N (AN), available soil P (AP), and available soil K (AK) were measured as described by Taylor and Francis (2012) (Carter et al., 2012).

### 2.5. Statistical Analyses

Data were examined for normality and homogeneity of distribution, and where required, we performed two-way ANOVA to test variance using cop rotation, soil fertilization, and year as fixed experimental factors. There were 2 or 3 soybean harvests in CSC and SCC crop rotations in 7 years. The effects of cop rotation, soil fertilization, and interactions on maize and soybean yield components and soil properties were calculated separately. Data in figures and tables are means calculated from three replicates. All statistical analyses of our data were performed with IBM SPSS Version 19 (Chicago, IL, USA) for Windows.

## 3. Results

### 3.1. Crop Yield and Quality Trait Response to Crop Rotation

After 7 years of different crop rotations and fertilizer treatments, the results showed a significant effect of crop rotation ($p \leq 0.01$) on both corn and soybean yield. The quality traits of soybean were significantly differentiated among different crop rotations, unlike soybean, where no significant effect on oil content was detected; a significant effect was found on the protein content of corn (Table 2). Our results showed that the yields of corn were significantly different under a CSC crop sequence compared to the CCC and SCC crop sequences, while the yield was not significantly different between the CCC and SCC crop sequences (Figure 1). The average yield of corn was lowest under SCC (11.21 Mg ha$^{-1}$) and greatest under CSC (11.81 Mg ha$^{-1}$) during the 7 years of the experiment (Table 2). The upscaled yields were about 0.6 Mg ha$^{-1}$ under CSC compared with SCC for corn. For corn yield component responses to rotation, a significant effect of rotation on plant height (PH) was observed among the treatment of rotations (Table S1). The average PH of corn was

lowest under CCC (227.5 cm) and greatest under CSC (235.9 cm) during the 7 years of the experiment. The relationships of PH and EL with yield for corn are shown in Figure 2. Both PH and EL of corn correlated significantly and positively with yield. However, soybean yields differed among the three crop sequences, with the highest variability under SCC and the lowest under the SSS crop sequence. The average yield of soybean under SCC was 0.21 Mg ha$^{-1}$ (about 9.7% higher) more than that of continuous monoculture soybean during the 7 years of the experiment. For soybean yield component responses to rotation, a significant effect of rotation on plant height (PH) and hundred grain weight (HW) was observed among the treatment of rotations (Table S2). The average PH of soybean was lowest under SSS (71.9 cm) and greatest under CSC (81.6 cm), while the average HW of soybean was lowest under SCC (21.2 g) and greatest under CSC (23.5 g) during the 7 years of the experiment. The relationships of PH and HW with yield for soybean are shown in Figure 2. Both PH and HW of soybean correlated significantly but negatively with yield.

**Table 2.** Effect of rotations and fertilization on yield (Mg ha$^{-1}$) and protein and oil contents (%) in a corn–soybean rotation during the 7 years after the initiation of the experiment (2014–2020). Different lowercase letters indicate significant differences at p < 0.05 using Fisher's least signifi-cant difference. Statistical significance at 5%, 1% and 0.1% denoted by *, ** and ***.

| Treatments | Soybean | | | Corn | | |
|---|---|---|---|---|---|---|
| | Yield | Protein Content | Oil Content | Yield | Protein Content | Oil Content |
| Rotation system | | | | | | |
| SSS | 2.16a | 41.44b | 19.87a | | | |
| SCC | 2.37c | 40.53a | 20.01b | 11.21a | 8.46c | 4.33 |
| CSC | 2.21b | 40.55a | 20.06b | 11.81b | 8.99b | 4.37 |
| CCC | | | | 11.28a | 8.31a | 4.32 |
| fertilization | | | | | | |
| Mineral fertilization | 2.21 | 40.71a | 19.69a | 11.38 | 8.36a | 4.22a |
| Biochar-based fertilization | 2.20 | 41.42b | 20.19b | 11.40 | 8.70b | 4.49b |
| Analysis of variance (F value) | | | | | | |
| rotation | 28.79 *** | 111.36 *** | 13.95 *** | 12.09 *** | 132.85 *** | 1.98 |
| fertilization | 0.01 | 160.40 *** | 151.73 *** | 0.15 | 140.87 *** | 60.54 *** |
| year × rotation | 8.95 *** | 9.31 *** | 7.09 *** | 4.87 *** | 17.98 *** | 0.37 |
| year × fertilization | 1.86 | 7.52 *** | 6.92 *** | 1.33 | 6.48 *** | 2.03 |
| rotation × fertilization | 0.64 | 4.95 * | 0.40 | 2.06 | 0.55 | 1.19 |
| year × rotation × fertilization | 0.23 | 5.95 ** | 3.79 * | 1.15 | 3.86 ** | 1.09 |

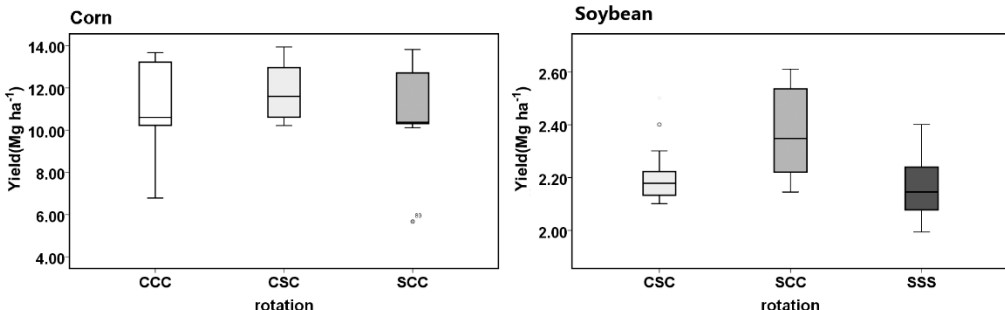

**Figure 1.** Boxplot of yield for corn and soybean under different crop rotations. Crop rotation treatments include continuous corn (CCC), corn−soybean−corn (CSC), soybean−corn−corn (SCC), and continuous soybean (SSS). ○ are outliers wich scores greater than 1.5 times the interquartile range are out of the boxplot.

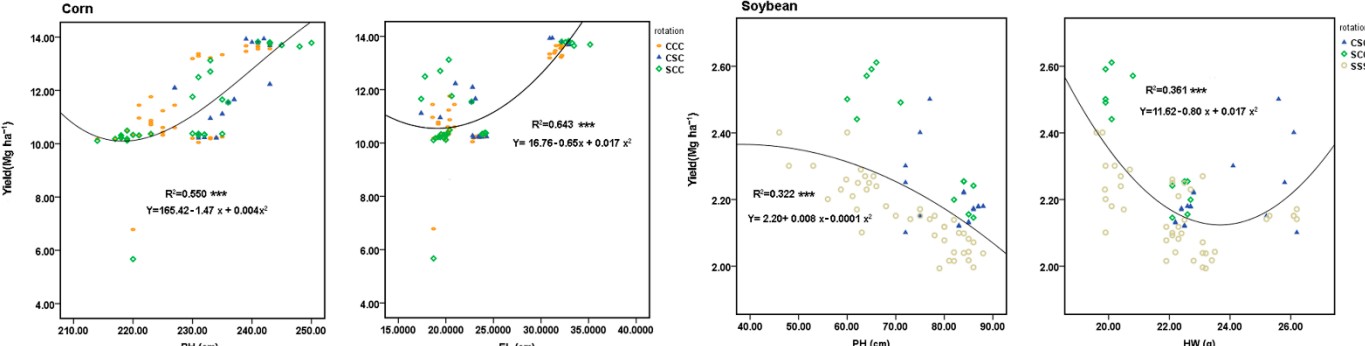

**Figure 2.** Relationship between yield and plant height (PH) or length of corn ear (LE) for corn. Relationship between yield and plant height (PH) or hundred grain weight (HW) for soybean. Different lowercase letters indicate significant differences at p < 0.05 using Fisher's least significant difference.

Our study detected significant differences in the protein content of corn among different crop sequences but no significant differences in oil content. The protein content of corn ranged from 8.31% to 8.99%, whereas a change tendency of protein content, which was CCC < SCC < CSC, was found among crop sequences (Table 2). Significant differences were found in the sequence of SSS compared to the sequences of SCC and CSC for both protein and oil content in soybean (Figure 3). Protein content was the highest (41.44%) and oil content (19.87%) the lowest for soybean in SSS among different crop sequences (Table 2).

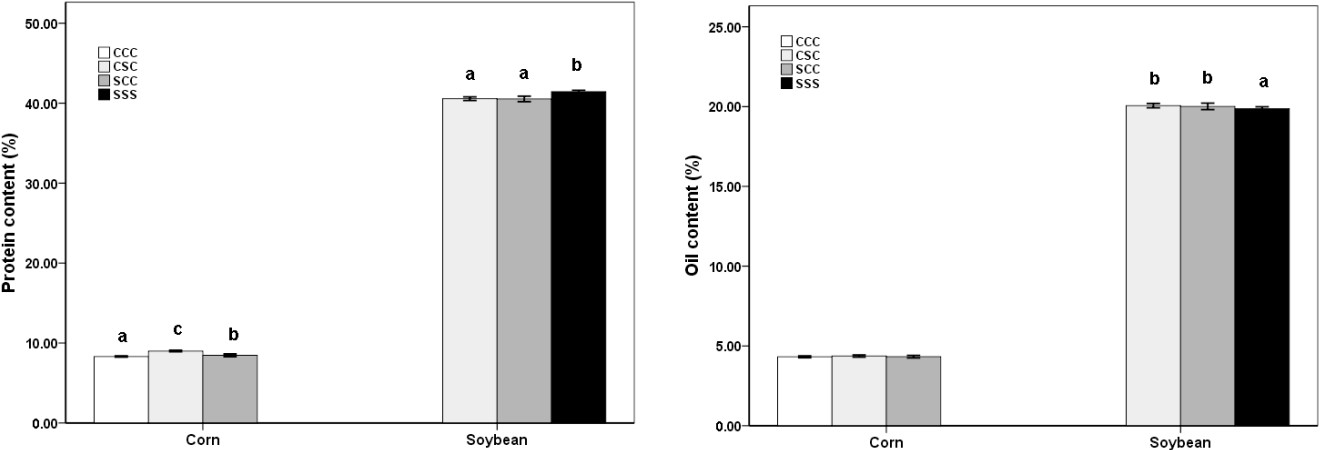

**Figure 3.** Protein and oil content of corn and soybean under different crop rotations. Different lowercase letters indicate significant differences at *p* < 0.05 using Fisher's least significant difference.

### 3.2. Crop Yield and Quality Trait Response to Fertilization Treatments

No differences in crop yield were detected between fertilization treatments, but a significant difference in the fertilization on both protein and oil content was observed for any crop (Table 2). Therefore, the grain yield of corn and soybean was further investigated for the fertilization effect over 7 years. The results revealed that the grain yield for corn showed a drop trend across years, yet an upward trend occurred for soybean, which ranked the highest in yield with $2.46 \pm 0.15$ Mg ha$^{-1}$ in 2018. Grain yields in 2014, the year of plot establishment, averaged $13.69 \pm 0.10$ Mg ha$^{-1}$ for corn and $2.10 \pm 0.03$ Mg ha$^{-1}$ for soybean. Grain yields before 2017 were not affected by biochar-based fertilization for corn. Biochar-based fertilization showed a lower corn grain yield than mineral fertilization in the 2017–2019 period, but not in 2020. However, the grain yield of soybean under biochar-based fertilization treatment showed higher than mineral fertilization in the first 4-year period (0.03 Mg ha$^{-1}$). but not thereafter. The grain yield was relatively lower in 2016 for both

corn (10.26 $\pm$ 0.83 Mg ha$^{-1}$) and soybean (2.02 $\pm$ 0.03 Mg ha$^{-1}$) among the seven years (Figure 4). This may have had a lot to do with the shortage of rainfall observed, relative to other years (Table 1). Interestingly, both protein and oil content were significantly greater with biochar-based fertilization than with mineral fertilization (Table 2). Averaged across the 7 years, the protein and oil content of soybean grown under biochar-based fertilization were greater (+0.71% and +0.50%) than those under mineral fertilization, whereas the upscaled contents of protein and oil were about +0.34% and +0.27% under biochar-based compared with mineral fertilization for corn.

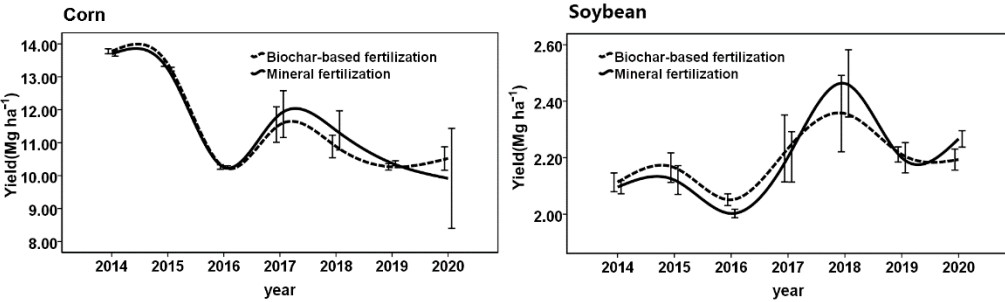

**Figure 4.** Effect of fertilization treatment on grain yield in a long-term study.

### 3.3. Effects of Crop Rotation on Soil Properties

Changes in percentage of total soil N (TN), P (TP), and K (TK) among all rotations are shown in Figure 5. The significant effect of crop rotation was found on the percentage of TN. The SSS crop sequence treatment illustrated the highest total soil N (TN) values at 0.18%, but the lowest total soil N (TN) value at 0.17% was found under the SCC crop sequence among all treatments. No significant effect of crop rotation was found on the percentage of either TP or TK. As a whole, the average percentage of TP and TK was about 0.15% and 0.24%, respectively. As for the total soil available N (AN), P (AP), and K (AK) content, the analysis showed that crop rotation treatments had significant effects on both AN and AK content (Figure 6). The SSS crop sequence treatment demonstrated the lowest AK content at 172.4 mg/kg and relatively lower AN content at 101.1 mg/kg, while the CCC crop sequence treatment illustrated the highest AK content at 185.8 mg/kg and also the highest AN content at 110.3 mg/kg. The lowest AK content occurred in the CSC crop sequence treatment at 96.4 mg/kg. Compared with the SSS crop sequence treatment, the CCC crop sequence treatment increased AN and AK content by 9.1% and 7.8%, respectively ($p < 0.05$). Other soil parameters such as AP, SOC contents, and PH value had no significant differences among all crop sequence treatments (Table S3).

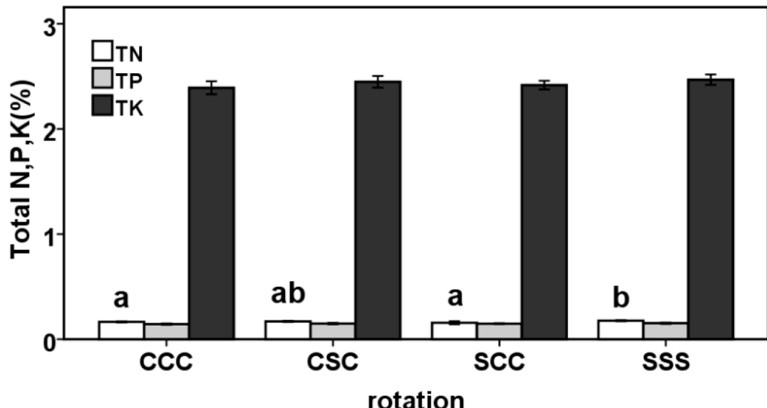

**Figure 5.** The effect of crop rotations on percentage of total soil N (TN), P (TP), and K (TK). Error bars represent standard deviations ($n = 3$). Different letters show significant differences among the treatments ($p < 0.05$).

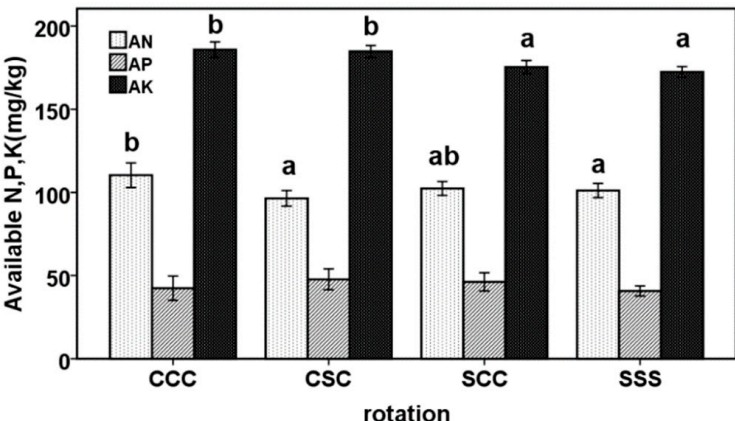

**Figure 6.** The effects of crop rotations on the total soil available N (AN), P (AP), and K (AK) content. Error bars represent standard deviations (*n* = 3). Different letters show significant differences among the treatments (*p* < 0.05).

*3.4. Performance in Crop Yield and SOC Content, Soil PH Value over Years*

The application of biochar-based fertilizers did not show a marked effect on the crop yield (Table 2), while as is the case in some long-term field studies, a significant effect on the yield of crop was found over the years (Figure 4). By averaging the fertilization treatment data, we observed that the yield of maize showed a declining pattern over the years, but an increasing tendency was found in the yield of soybean. The mean maximum seed yield of corn was found in 2014 (13.73 Mg ha$^{-1}$), while the minimum seed yield was found in 2020 (10.22 Mg ha$^{-1}$). In general, no obvious trend of maize yield was found in difference between fertilization treatments over the years. However, the mean maximum seed yield of soybean was noted in 2018 (2.41 Mg ha$^{-1}$), followed by 2020 (2.23 Mg ha$^{-1}$), while the minimum seed yield was found in 2016 (2.03 Mg ha$^{-1}$). In addition, the biochar-based fertilization treatment displayed lower yields by 4.3% than the mineral fertilization treatment in 2018.

No significant effect of biochar-based fertilization was found on SOC content compared with mineral fertilization treatment. The accumulation of organic matter in SOC increased over years, growing to a relatively high level of 17.4 g/kg in 2020 from 14.8 g/kg in 2014 (Figure 7). However, changes in PH value had the opposite trend compared to soil SOC content over the years (Figure 8). The PH value decreased from 8.8 to 8.0 in 2020.

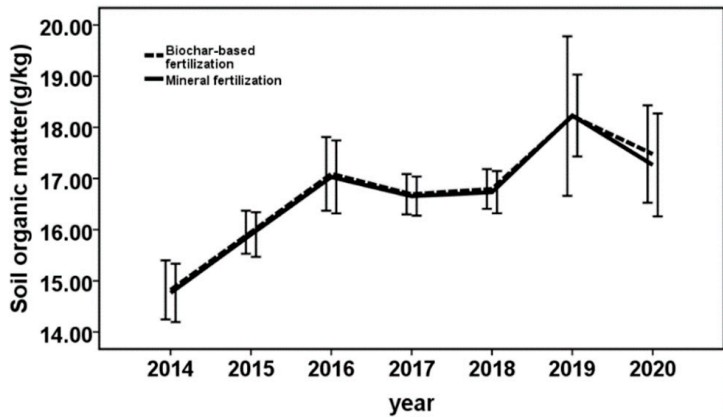

**Figure 7.** Soil organic matter (SOC) variation for fertilization treatment over 7 years.

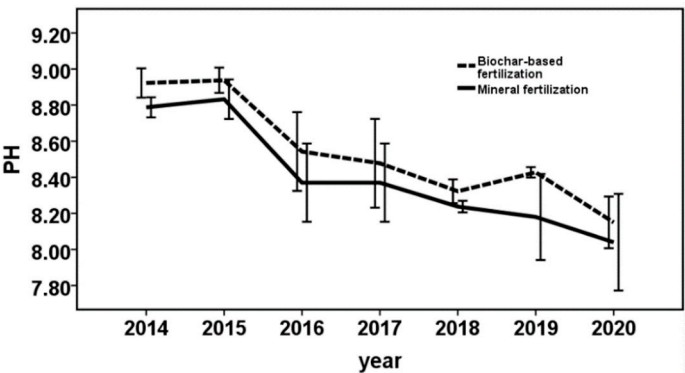

**Figure 8.** PH value (PH) variation for fertilization treatment over 7 years.

## 4. Discussion

### 4.1. Long-Term Rotation Had Positive Effect on Crop Yield

Generally higher yields of crop grown in the rotations occurred often compared with continuous monoculture [31–33]. In our study, corn showed a small yield increase of 4.7% when grown in the crop sequence of CSC compared with continuous monoculture. The relative yield advantage of corn grown in CS compared with CC has been well-documented [15,31]. Seifert et al., (2017) found a yield increase of 4.3% for corn in 2-year rotation compared with continuous monoculture between 2007 and 2012 in the US Midwest [34]. Such yield improvements have been attributed to reducing pest pressure [35] and increasing the availability of N, which was symbiotically fixed in previous soybean crop [32]. However, the CCC crop sequence treatment illustrated the highest AN content at 110.3 mg/kg in this study. All yield components measured in this study were significantly related to corn yield. The highest correlation (r = 0.776) was observed between yield and the length of corn ear (LE), followed by plant height (PH) (r = 0.720). Averaged across the 7 years, as PH or LE declined, the yield of corn tended to decline; moreover, the yield response to rotations tended to increase under CSC compared with the CCC crop sequence. For soybean yield, the treatments of crop rotation had a better performance than continuous soybean during the seven years in our study, with the yield of soybean in the sequences of SCC and CSC being greater than SSS by an average 10.2% and 2.8%, respectively. The lower yields for SSS may be attributed to more rooting constraints and greater depletion of available nutrients such as N, P, and K compared with crop rotation [36], but also to decreasing microbial populations which are beneficial for soil, while increasing the relative abundance of pathogenic fungi [37]. Crop yield may be affected not only by the availability of soil nutrient but also by climate conditions, including rainfall and temperature. Some researchers have also reported that crop yield was influenced by climate factors such as spatial and temporal variation based on observational weather data of air temperature and precipitation from multistations for long-term experiments [38]. Corn yield under corn–soybean rotation was found to be positively related to rainfall in the anthesis and kernel-filling periods [39]. In the current study, a similar rule regarding rainfall was obtained from May to September, where both corn and soybean yield appeared to drop when precipitation decreased from 410.9 mm in 2015 to 282.2 mm in 2016. The longer term of the study combined with rainfall, temperature, and other factors could be appropriate in understanding how climate may affect crop yield. For yield components measured in this study, plant height (PH) and hundred grain weight (HW) were significantly related to soybean yield. Averaged across years, as PH or HW increased, soybean yield tended to decline; moreover, the yield response to rotations tended to increase under SCC compared with the SSS crop sequence.

### 4.2. Crop Yield and Quality Trait Response to Fertilization Treatments

Long-term experimentation has been proven to be a valuable approach for quantifying the interaction between crop rotations and soil properties or nitrogen dynamics [40–42].

However, our results showed that diverse rotations had a positive influence on total soil N (TN) compared with corn monoculture, but a negative influence compared with monoculture soybean. For instance, greater TN concentration was observed under the SSS crop sequence, which was contrary with the findings of Van Eerd et al. [43] and Congreves et al. [42]. The content of SOC is one of the most important main parameters that influence soil physical properties. The SOC values were lower in short crop–soybean–corn rotation compared with corn monoculture [44,45]. Greater accumulation of SOC under more complex rotations was observed compared with that under short rotation or monoculture [46]. We, however, did not test significant differences of SOC among all crop sequence treatments.

Crop rotation may also affect soil properties, including soil carbon stocks and nutrient availability [47]. Research has shown that crop rotation changed the microbial community structure, which may influence the nutrient content of soil, such as available nitrogen and soil organic matter [48]. In this study, the results revealed that crop rotation treatments had significant effects on both available nitrogen (AN) and available (AK) content. The SSS crop sequence treatment demonstrated the lowest AK content and a relatively lower AN content. However, no significant differences in available nitrogen (AN) were found between conventional rotation and extended rotation in previous research [49]. The correlation between change in soil properties and environmental factors is still complicated.

No differences in crop yield were detected between fertilization treatments in this study. Muhammad Arifa et al., (2014) found that there was a significant effect of biochar on the biological yield, plant height, and leaf area index (LAI) of both maize and wheat [26]. Deborah et al., (2018) found that the levels of soil carbon increased (8–115%) with the increase in biochar application rate (0–90 Mg ha$^{-1}$) for both corn monoculture and maize–soybean rotations, but only a small impact was found on corn yields (–2.6 to 0.6%) [50]. However, our study detected a positive effect of biochar-based fertilizer application in combination with mineral fertilization on corn and soybean quality properties. Biochar-based fertilization increased both the protein and oil content of two crops compared with no biochar applied. Consistent with previous findings, Zahra reported that the effect of biochar treatments on the oil content ($p \leq 0.01$) as well as protein content of soybean was significant [51]. It has been found that biochar application affected the carbon and nitrogen presence in the soil, and it reduced the mineralization process, thus making the nutrients available to the plants [52]. We did not detect any adverse effects of soil pH changes induced by biochar application on both crops yields [53,54]. Our results suggest that the soil N effects are responsible for yield benefits observed in the diversified rotation.

## 5. Conclusions

Crop rotations have been reported to have many beneficial effects on yields and soil physical, chemical, and biological properties. We investigated different crop rotations on yield, quality traits, and soil properties under fertilization treatments over 7 years. Our results demonstrated that crop rotation had a significant positive impact on all crop yield, and protein and oil content of soybean. The greatest average yield of corn (11.81 Mg ha$^{-1}$) was detected in the CSC crop sequence, while the greatest average yield of soybean (2.37 Mg ha$^{-1}$) was detected in the SCC crop sequence during the 7 years of the experiment. Unlike soybean, a significant effect was found on the protein content of corn only in long-term corn–soybean rotation treatments. For crop yield component responses to rotation, a significant effect of rotation on plant height (PH) was observed among the treatment of rotations. However, no differences in crop yield were detected between fertilization treatments, but a significant positive effect of biochar-based fertilization on both protein and oil content was observed for any crop. A significant effect of crop rotation was found on the percentage of TN, AN, and AK content. Our results suggested that the rotation system seemed to be a more important factor improving grain yield and quality traits. Therefore, future research will be needed to further understand the rotation effects on soil properties as well as subsequent crop growth and yield.

**Supplementary Materials:** The following supporting information can be downloaded at: https://www.mdpi.com/article/10.3390/agronomy12102554/s1, Table S1: Effect of rotations on all measured parameters of corn; Table S2: Correlation for soybean; Table S3: Effect of rotations and treatment on all soil parameters.

**Author Contributions:** Conceptualization, Y.L. (Yuehui Liu) and Y.B.; methodology, D.H. and M.L.; software, L.W. (Ling Wang) and J.L.; validation, D.Z., Z.W. and Z.Z.; formal analysis, H.S.; investigation, J.W. and S.D.; resources, B.M.; data curation, C.F.; writing—original draft preparation, M.Y. and G.Y.; writing—review and editing, Y.B.; visualization, L.W. (Lianxia Wang), W.L. (Wenwei Liang) and W.L. (Wei Li); project administration, Y.L. (Yongcai Lai); funding acquisition, Y.B. All authors have read and agreed to the published version of the manuscript.

**Funding:** This project is supported in part by the Scientific research business cost project of Heilongjiang provincial scientific research institutes "Research and application of key techniques for improving yield and quality of soybean based on crop rotation", Grant Number CZKYF2020C004; China Agriculture Research System of MOF and MARA, Grant Number CARS-04; Scientific research business cost project of Heilongjiang provincial scientific research institutes, Grant Number CZKYF2022-1-B020; and Scientific and technological research project of Heilongjiang Academy of Agricultural Sciences, Grant Number 2021YYYF009.

**Conflicts of Interest:** The authors declare no conflict of interest.

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
