# Peer review of "Long-Term Corn–Soybean Rotation and Soil Fertilization: Impacts on Yield and Agronomic Traits"

_agronomy, doi:10.3390/agronomy12102554_

Round 1

Reviewer 1 Report

The authors demonstrated the impact of long-duration crop rotation and soil fertilization in corn and soybean plants in terms of productivity and agronomic characteristics. The article presentation is good. The article can be acceptable after substantial revision.

Line 31:  .........significant effect of crop rotation was found on the percentage of TN, AN and AK content. please write the % value

Line 25, 155, 156: .................Mg ha-1 or mg ha-1, plz check the entire MS

Line 42: Plz correct the plants scientific name 

Line 91: Please modify Table 1 presentation

Figure 1, 4: Plz correct the Y-axis caption

Result and discussion section needs improvement

Note: 1. Improve the English language entire the MS, some sentences meaning is not clear

2. Plz check the reference style as per the journal requirement

Reviewer 2 Report

In attached is revision of manuscript. 

Reviewer 3 Report

The manuscript is focused on the long-term effects of crop rotation and fertilization on yield and other agronomic traits. I appreciate the large amount of work that must have done into the experiment for this long-term study, however mainly methods, data analyses and their presentation must be improved. I have following recommendations mainly for methods and results sections:

Introduction:

L69-74: The part providing background about the effect of biochar and other carbon amendments on soil fertility and crop yield must be extended.

Methods:

L85: You should add the altitude of experimental site and include that location is in China.

L87-88: “The average temperature in summer…” You should specified the time period or use annual average.

L88-89: You should better determine the soil type, e.g. according to FAO soil groups.

Table 1: I recommend to use percentage differences in temperature and precipitation in comparison with long-term averages. Maybe the graphical presentation in figure will be better.

L89-90: Add how you measured air temperature and precipitation.

L96-98: I understand well that in the case of rotation CSC were following crops planted during 7 years (CSC CSC C) and in the case SCC (SCC SCC S)? It means 2 or 3 soybean harvests in these two crop rotations during 7 years, respectively? It should be mentioned in the description of statistical analysis.

L101 x L111: Unify the units (hm2 vs ha).

L100-103: Add the dates of application of individual doses of fertilization.

L109: Use “sowing” instead of “planting” in the whole text.

L115: What type of herbicides were used?

L130: How many soil samples were taken for each plot? Add the diameter of the auger used.

L135-136: You have to specified individual methods used in the laboratory.

L139: You write “We pooled the data from cropping cycles and analysed the data using two-way analysis of variance test using cop rotation and soil fertilization as fixed experimental factors”, but in Table 2 you show interactions with year. Is it different analysis?

In addition, add the description of correlation analyses used in figure 2.

Results:

L156-158: You write “For corn yield component responses to rotation, a significant effect of rotation on the plant height (PH) was observed among the treatment of rotations”, however no statistical analysis for other parameters is shown.

I recommend to add the tables with ANOVAs for all measured parameters, maybe as a supplement material.

L160: You link to Fig. 2-1 and later to Fig. 2-2, however the figure is not described into two parts.

L160-161: You write “Both PH and EL of corn correlated significantly but negatively with yield”, but according to figure it seems that correlated positively.

In addition, in Fig. 2 there are not described significances and also any description why you use non-linear fitting.

L161-162: “However, soybean yields differed among three crop sequence with the highest variability under SCC and lowest under SSS crop sequence.” According to Fig. 1, the lowest variability is under CSC, under SSS there is the lowest value (mean).

L163: Change “0.21” to “11.21”. And what the percentage value 9.7% means?

L164-166: “For soybean yield component responses to rotation, a significant effect of rotation on the plant height (PH) and hundred grain weight (HW) was observed among the treatment of rotations”. Same as mentioned above - no statistical analysis for other measured parameters.

Table 2: Add to the heading of table “Results of two-way ANOVA analysis (F values; *P ≤ 0.05, **P ≤ 0.01, ***P ≤ 0.001). Different letters denote statistically significant differences between means from individual treatments with p ≤ 0.05.

In addition, the first part of the Table 2 (rotation system) is doubled by Figs. 1 and 3. You should choose between table and figure.

Figure 4: here you show effect of fertilization on yield during whole experiment (not significant difference). Did you try to create a similar figure for protein or oil contents? Or for individual rotation system? Maybe you find any effect of fertilization in any specific rotation system when you have some significant interactions between year x rotation x fertilization (for protein or oil contents).

In addition, when you described differences between years (e.g. L196-212), did you try to make correlation analyses between yield or other agronomic parameter and climate variables (air temperature or precipitation)?

L216-231: ANOVA for soil parameters is missing. I recommend to create Figs. 5 and 6 separately for individual parameter and show both fertilization treatments. You could find some interesting interaction between fertilization and crop rotation.

Fig. 7: When you did not find any effect of fertilization on SOC for all rotation treatments together, there is no any difference within individual crop rotation system? Unfortunately, no ANOVA for soil parameters is available.

In my opinion, only when the results section will be improved according to mentioned recommendations, the discussion section and also conclusions of the paper could be evaluated.

Reviewer 4 Report

Abstract

- Underscore the scientific value-added to your paper in your abstract. Your abstract should clearly state the essence of the problem you are addressing, what you did and what you found and recommend. That will help a prospective reader of the abstract to decide if they wish to read the entire article.

- Please should the treatments in this section.

- Lines 17-20: This sentence is too long. Please revise it.

- Line 20: Please add the country name after Heilongjiang province.

- Line 21: On the crop yield …

- Lines 22-23: The sentence of ‘We used 7-yr average crop yield variability …’ is repetitive and should be deleted.

- Line 25: What is the CSC?

- Line 25: The authors noted that ‘The corn yield was enhanced by 0.6 Mg ha−1 in CSC crop sequence’. Please add the increasing percentage.

- Line 26: Please add the increasing percentage for soybean yield, too.

- Line 25-27: The two sentences should be mixed.

- Line 27: The sentence ‘However, our study detected negative effect of crop rotations …’ is unclear.

Line 29: The authors noted that ‘No differences in crop yield were detected between fertilization treatments …’. Which fertilization treatments?

- Lines 31-32: What is the KN, AN and AK?

- Based on the obtained results, what is the best recommendation?

Introduction

- The linkage between paragraphs is missed.

- In the first paragraph the authors should be added more information’s about the negative impacts of intensive agriculture systems.

Lines 42-45: Reference?

- Lines 46-47: Which studies? Please explain.

- Line 50: ‘Kazula (2018) reported’. Please change the citation format based on the Journal format.

- Line 51: what is the CSW, CS and CC?

- Line 53-54: The sentence is not clear.

- Justify novelty in Introduction and Discussion.

- Please subject the manuscript to review made by English Native speaker.

- Line 61: ‘Martens et al. (2006)’ should be modified based on the journal format.

- Lines 77-81: The novelty of this study is missed.

Materials and methods

- Line 95: ‘on yield and agronomic traits of crop’ Which crop or crops?

- Why the field experiment was arranged as a split-plot design?

- Lines 147-151: The sentence is not clear.

Results

- In table and figure captions, the authors should completely explain the abbreviations words.

Table 2: Please add the units of each trait.

Table 2: Why the interaction of year × fertilization in the protein, oil content of soybean and yield of corn was significant? The amount and type of fertilizers different in growing years?

- Please add the increasing or decreasing percentage between treatments?

- Given that the study presents a long list of abbreviations, I suggest adding a “glossary” table at the end of the paper as it will aid the readers to learn about the concepts/terms that they are about to study.

Discussion

- Line 263: ‘Mumber is wrong’.

- This section is not really discussion. In this section the authors listed the previous studies results. Please add the main reasons for increasing or decreasing the measured traits.

- Line 319: The citation format is not wrong.

- The novelty of study is missed. Previously research and books reported the effects of rotation crops in comparison with plants continuous cropping. What is the novelty of this study?

Conclusion

- This section is repetitive and should be rewritten.

- Please make sure your conclusions' section underscores the scientific value-added of your paper, and/or the applicability of your findings/results. Highlight the novelty of your study.

References

- The reported references are not adjusted according to the journal format.

Round 2

Reviewer 1 Report

The authors incorporated all corrections nicely.  References 53 and 54 are missing in the text, and Ref. 54 is incomplete in the reference list. Plz correct before acceptance.

Reviewer 3 Report

The authors made many changes in the manuscript, however some comments and recommendations mentioned previously were not answered. See below.

I would like to see answers of authors for all my comments where is not written “Ok”.

Introduction:

L69-74: The part providing background about the effect of biochar and other carbon amendments on soil fertility and crop yield must be extended.

Ok.

Methods:

L85: You should add the altitude of experimental site and include that location is in China.

No change.

L87-88: “The average temperature in summer…” You should specified the time period or use annual average.

No change.

L88-89: You should better determine the soil type, e.g. according to FAO soil groups. Ok.

Table 1: I recommend to use percentage differences in temperature and precipitation in comparison with long-term averages. Maybe the graphical presentation in figure will be better. No any change. I don´t see reason of this table. I still think that percentage differences from long-term average will better show difference between years or some weather extremes.

L89-90: Add how you measured air temperature and precipitation.

No change.

L96-98: I understand well that in the case of rotation CSC were following crops planted during 7 years (CSC CSC C) and in the case SCC (SCC SCC S)? It means 2 or 3 soybean harvests in these two crop rotations during 7 years, respectively? It should be mentioned in the description of statistical analysis.

I do not see any answer to my question or some improvement of this in the text.

L101 x L111: Unify the units (hm2 vs ha). Ok.

L100-103: Add the dates of application of individual doses of fertilization. Ok.

L109: Use “sowing” instead of “planting” in the whole text. Ok.

L115: What type of herbicides were used? Ok.

L130: How many soil samples were taken for each plot? Add the diameter of the auger used. Ok.

L135-136: You have to specified individual methods used in the laboratory. Ok.

L139: You write “We pooled the data from cropping cycles and analysed the data using two-way analysis of variance test using cop rotation and soil fertilization as fixed experimental factors”, but in Table 2 you show interactions with year. Is it different analysis?

Ok.

In addition, add the description of correlation analyses used in figure 2.

No change. There is no description of correlation analyses in Methods or in Figures.

Results:

L156-158: You write “For corn yield component responses to rotation, a significant effect of rotation on the plant height (PH) was observed among the treatment of rotations”, however no statistical analysis for other parameters is shown.

No change. I see statistical analyses only for yield, protein content and oil content.

I recommend to add the tables with ANOVAs for all measured parameters, maybe as a supplement material.

L160: You link to Fig. 2-1 and later to Fig. 2-2, however the figure is not described into two parts. Ok.

L160-161: You write “Both PH and EL of corn correlated significantly but negatively with yield”, but according to figure it seems that correlated positively. Ok.

In addition, in Fig. 2 there are not described significances and also any description why you use non-linear fitting. Ok.

L161-162: “However, soybean yields differed among three crop sequence with the highest variability under SCC and lowest under SSS crop sequence.” According to Fig. 1, the lowest variability is under CSC, under SSS there is the lowest value (mean). Ok.

L163: Change “0.21” to “11.21”. And what the percentage value 9.7% means?

L230: The average yield of soybean under SCC is 2.37? Not 0.21. Am I right?

L164-166: “For soybean yield component responses to rotation, a significant effect of rotation on the plant height (PH) and hundred grain weight (HW) was observed among the treatment of rotations”. Same as mentioned above - no statistical analysis for other measured parameters.

No change. I see statistical analyses only for yield, protein content and oil content. You can not comment results about plant height or hundred grain weight when the statistical analyses of these parameters are not presented.

Table 2: Add to the heading of table “Results of two-way ANOVA analysis (F values; *P ≤ 0.05, **P ≤ 0.01, ***P ≤ 0.001). Different letters denote statistically significant differences between means from individual treatments with p ≤ 0.05.

No any change.

In addition, the first part of the Table 2 (rotation system) is doubled by Figs. 1 and 3. You should choose between table and figure.

No any change.

Figure 4: here you show effect of fertilization on yield during whole experiment (not significant difference). Did you try to create a similar figure for protein or oil contents? Or for individual rotation system? Maybe you find any effect of fertilization in any specific rotation system when you have some significant interactions between year x rotation x fertilization (for protein or oil contents).

I would like to see answer to this previously mentioned comment.

In addition, when you described differences between years (e.g. L196-212), did you try to make correlation analyses between yield or other agronomic parameter and climate variables (air temperature or precipitation)?

I would like to see answer to this previously mentioned comment.

L216-231: ANOVA for soil parameters is missing. I recommend to create Figs. 5 and 6 separately for individual parameter and show both fertilization treatments. You could find some interesting interaction between fertilization and crop rotation.

I don´t see any change in figures. I would like to see answer to this previously mentioned comment.

Fig. 7: When you did not find any effect of fertilization on SOC for all rotation treatments together, there is no any difference within individual crop rotation system? Unfortunately, no ANOVA for soil parameters is available.

I would like to see answer to this previously mentioned comment.

Reviewer 4 Report

- The experimental treatments were not added in abstract.

- Lines 41-44: The two sentences have the same meaning. Please revise this section.

- Justify the novelty in introduction and discussion. 

- The analysis method is not true. The authors noted that ‘Moreover considering the field or time consuming. the field experiment was arranged as a split-plot design’. The experimental factors of this study were different cropping patterns and fertilizer application.

- In discussion section the authors add repetitive sentences from results section and after that compare the obtained results with other published results. the novelty is missed.

Previously research and books reported the effects of rotation crops in comparison with plants continuous cropping. What is the novelty of this study?
